# Exposure to natural hazard events unassociated with policy change for improved disaster risk reduction

Daniel Nohrstedt [1,2✉], Maurizio Mazzoleni[2,3], Charles F. Parker [1,2] & Giuliano Di Baldassarre [2,3]

Natural hazard events provide opportunities for policy change to enhance disaster risk reduction (DRR), yet it remains unclear whether these events actually fulfill this transformative role around the world. Here, we investigate relationships between the frequency (number of events) and severity (fatalities, economic losses, and affected people) of natural hazards and DRR policy change in 85 countries over eight years. Our results show that frequency and severity factors are generally unassociated with improved DRR policy when controlling for income-levels, differences in starting policy values, and hazard event types. This is a robust result that accounts for event frequency and different hazard severity indicators, four baseline periods estimating hazard impacts, and multiple policy indicators. Although we show that natural hazards are unassociated with improved DRR policy globally, the study unveils variability in policy progress between countries experiencing similar levels of hazard frequency and severity.

[1] Department of Government, Uppsala University, 75120 Uppsala, Sweden. [2] Centre of Natural Hazards and Disaster Science (CNDS), c/o Department of Earth Sciences, Uppsala University, 75236 Uppsala, Sweden. [3] Department of Earth Sciences, Uppsala University, 75236 Uppsala, Sweden. ✉email: daniel.nohrstedt@statsvet.uu.se

Floods, wildfires, storms, and other natural hazard events interact with societal vulnerability to cause human casualties, property damage, and economic loss. Reducing disaster losses is directly linked to the achievement of the sustainable development goals (SDGs), including targets to build resilience and reduce vulnerability to climate-related extreme events (SDG1, target 1.5) and to reduce losses from these events in terms of deaths, people affected, and economic impacts (SDG11, target 11.5)[1,2]. While much research has been directed at understanding variability in hazard losses related to income, geography, and level of democracy[3–5], few studies have yet examined natural hazards as drivers of policy change to reduce disaster risk around the world[6].

When the Hyogo Framework for Action (HFA) was adopted in 2005, it marked an important milestone in the development of an international policy regime to support countries in their efforts to reduce the risk from natural hazards. Coordinated through the United Nations Office for Disaster Risk Reduction (UNISDR, currently UNDRR), the HFA for the first time provided a set of attainable policy actions for enhancing legal and institutional resources and creating enabling environments for stakeholder collaboration in the pursuit of disaster risk reduction (DRR)[7]. Generally, such policy directives have a low chance of succeeding because of well-known rigidities characterizing policymaking, including inertia, path dependency, and incrementalism[8]. Due to these hurdles, the dominant theoretical explanation for policy change has been the occurrence of external shocks, including extreme natural hazard events.

Cataclysmic events can spur policy change by raising public consciousness, redistributing political resources among policy actors, and providing impetus and legitimacy to governmental action[9]. Policy change has been attributed to repeated events that gradually build pressure for reform or high-magnitude events whose social and economic impacts cannot be ignored by policymakers[10,11]. This perspective, here labeled the "disaster-reform hypothesis," is well established in several research fields, figuring prominently in public policy[12], environmental management[13], resilience[14], and adaptation, transformation, and transitions[15,16] in the pursuit of sustainable development[17,18]. Other studies suggest that extreme hazard events are unlikely to trigger major policy change. For instance, research[19,20] has shown that countries regularly exposed to large-scale natural hazard events tend to devote resources to recovery at the cost of developing proactive DRR policies. Hazard events can also spark political contestation and become portrayed as policy failures, encumbering impartial diagnosis, learning, and system improvement, and can incentivize political and bureaucratic leaders to restore order, which may reaffirm pre-existing policies[21].

Despite these competing insights and expectations, systematic empirical research investigating the policy effects of natural hazards has been sparse. Prior work primarily draws from single or small-$N$ case studies and has concluded that some hazard events trigger significant policy change while others reinforce the status quo and even increase the vulnerability of communities[22–24]. A few systematic studies[25–29] have analyzed the influence of past natural hazards on damage reduction as a proxy for adaptation and also reported mixed results. Comparative studies have documented socio-political effects of natural hazards, but this work is limited to a few phenomena, including civil and political unrest[11,30], armed conflict[31], political regime change[23], elections[32], community memory[33], and climate change discourse[34]. However, while the impact of natural hazard events on policymaking, specifically policy change, is extensively theorized and contested in the scientific literature, conclusive empirical evidence is lacking regarding the relationship from a global perspective. This lack of knowledge is unfortunate, given

that the frequency and intensity of some of these events are likely to increase in a changing climate[20].

This study aims to empirically explore if frequency (number of events) and severity factors (fatalities, people affected, and economic losses) of natural hazards influence national-level changes in DRR policy. To test this, we combine data on 85 countries collected from the HFA regime with disaster event data and explore whether the frequency or severity of natural hazard events have influenced national-level DRR policymaking worldwide.

Data were retrieved from four evaluation cycles in the HFA covering the 2007–2015 period, focusing on countries' self-reported progress on a scale from 1 (minor achievement) to 5 (comprehensive achievement), in implementing 22 key activities detailed by the HFA. Activities were listed under five priority for action (PFA) areas, specifying measures and principles for enhancing disaster resilience (Supplementary Table 1). Since the data is self-reported, we performed a validation test for two selected countries with the highest policy change scores (Swaziland and Chile), including assessing their HFA progress scores and level of policy change (see "Methods"; Supplementary Results). DRR policy change was measured as changes in the average of all five PFAs combined and for each PFA separately. These data were combined with information retrieved from the *International Disaster Database* (EM-DAT)[35], which is the most widely used global natural hazard events database capturing the date, location, and impacts in terms of economic losses and the number of persons killed, injured, and affected[36]. Given the observation that the nature and severity of past hazard events can influence policy learning[37], hazard frequency and severity measures were normalized in relation to long-term country averages in four baseline periods. The analysis also considered that the frequency and severity of natural hazard events are relative to countries' experience of past events at different time scales. Finally, we controlled for income levels, differences in PFA values reported in the first two evaluation cycles, and hazard event types as potential confounding variables.

## Results

The HFA data provide a valuable opportunity to measure policy change through clearly defined starting conditions (average progress scores in the first two HFA evaluation cycles) based on concrete policy efforts for longitudinal cross-country comparison[38]. For each country, DRR policy change was measured as the difference between the average HFA progress scores in the five PFAs between two main periods in 2007–2011 and 2011–2015 (Eqs. (1) and (2), "Methods").

**DRR policy change and relationships with natural hazard events**. Regarding patterns of DRR policy change, measured by changes in average PFA values, we find that policies in PFA3, which cover the use of knowledge, innovation, and education, changed the most over the study period (0.25 on average). The second largest change occurred in PFA4, which addresses the reduction of underlying risk factors (0.22) followed by PFA2, which deals with disaster risk and early warning (0.19). Policies in PFA1, which concern DRR as a national and local priority, including institutional conditions for implementation, and, PFA5, which focuses on measures to strengthen preparedness, changed the least (0.15 and 0.16, respectively). The majority of the countries ($n = 63$, 74%) reported positive policy change (a shift from lower progress scores in 2007–2011 to higher scores in 2011–2015—defined here as "improved" DRR policy), 6 (7%) were defined as status quo (same scores in both periods), whereas

17 (20%) had negative changes (a decline from higher to lower scores).

Next, we compared these policy changes in the 2011–2015 period with natural hazard event frequency and severity measures in the 2007–2011 period. Figure 1 plots policy changes by country in relation to fatalities (Fig. 1a), affected people (Fig. 1b), economic losses (Fig. 1c), and the number of events (Fig. 1d). Average PFA changes refer to aggregated measures of change in the five PFAs in the 2007–2015 period (Eqs. (1) and (2), "Methods"). Severity and frequency measures are normalized in relation to the 30-year baseline period, i.e., estimating whether frequency and severity measures in the study period were above or below average in a 30-year period between 1980 and 2011. Here severity and frequency values >1 ($x$-axis) indicate that countries in the 2007–2011 period experienced hazard events that were more severe and/or more frequent compared to average frequency/ severity in the 1980–2011 period (see "Methods"). Figure 1 classifies countries by income levels to account for possible wealth effects, particularly the assertion that lower-income countries generally have less ability to adapt to extreme events[39–41]. Based on these results, we finally highlight pairs of two candidate cases per factor for future comparison and in-depth study.

We find that both the level and variability of policy change— independent of exposure to natural hazard events—is lower on average among high-income countries (average PFA change = 0.12, standard deviation = 0.23) compared to countries at lower-income levels. Although there are examples of outlier cases, represented by high-income countries with high average PFA changes (for example, Chile, average PFA change = 0.85, Fig. 1) as well as low-income countries with relatively small or negative average PFA changes (for example, Togo, average PFA change = −1.21, Fig. 1), Fig. 1 shows that most high-income countries are located close to the status quo threshold indicating policy stability (i.e., average PFA change = 0, Fig. 1a–d). This stands in contrast to countries at lower-income levels, which displayed more substantial PFA changes on average and greater variability in terms of standard deviation values (low income = 0.19, SD = 0.56; lower–middle income = 0.26, SD = 0.48; upper–middle income = 0.17. SD = 0.45). However, given that policy change is measured as the difference between progress scores on the five-point scale, these differences between income levels are relatively negligible.

Results reported in Fig. 1 rely on a relatively simple measure of DRR policy change, based on the average progress of all five PFAs combined. Furthermore, these results are based on the 30-year baseline for normalizing frequency and severity and thus do not show results based on other baselines. To explore potential differences across individual PFAs, Fig. 2 relates average changes in each PFA to different levels of hazard severity (Fig. 2a–c) and frequency (Fig. 2d) on an aggregated level. We also show results for these policy changes in relation to all four baseline periods (1970 [40 years], 1980 [30 years], 1990 [20 years], and 2000 [10 years]). By comparing results across the four baselines, we are able to examine whether policy impacts of natural hazard events depend on different measures of frequency and severity, respectively. Negative difference values in Fig. 2 ($y$-axis) indicate that countries exposed to a greater number or more severe events relative to the long-term country averages reported lower levels of policy change than countries with fewer and less severe events. Conversely, positive difference values indicate higher policy change levels in the former group of countries compared to the latter (Eq. (4), "Methods").

Figure 2 shows, contrary to the disaster-reform hypothesis, that an increase in hazard frequency and severity is generally followed by less policy change. To the extent there was change, it tended to be negative. In fact, countries experiencing more frequent and severe hazard events reported lower levels of DRR policy change compared to countries with fewer and less severe events. This result is also depicted in Fig. 1, which illustrates that several countries with high frequency (Fig. 1d) and severity scores (Fig. 1a–c) had low or even negative policy change values. For instance, we find several examples of countries that experienced economic losses higher than the long-term average (Fig. 1c) but reported PFA change close to 0, among them Mexico, Vietnam, Australia, Cuba, Japan, Romania, Malaysia, New Zealand, and Thailand. As shown in Fig. 2, the pattern is relatively similar across the five PFAs and the four baseline periods, yet some interesting variations can be noted.

The five PFAs display relatively similar differences between the two groups of countries in relation to fatalities (Fig. 2a), affected people (Fig. 2b), and the number of events (Fig. 2d). In comparison, there is more variance between the five PFAs in relation to economic losses (Fig. 2c). It is shown here that hazard events causing greater economic losses are, in fact, associated with lower levels of policy change compared with events causing less economic loss. This suggests that the positive relationship between economic losses and policy change reported in some case studies[37,42] does not apply globally when focusing on DRR policy. The pattern regarding policy impacts of economic losses, however, is most pronounced for PFA3 involving the use of knowledge, innovation, and education in support of safety and resilience (difference values below −0.2 for all baseline periods, Fig. 2c). This is in contrast to policies in PFA1 (DRR as a national and local priority) and PFA5 (measures to strengthen preparedness), which changed more after hazard events with greater economic losses compared to events generating less economic damage. Results are similar for affected people (Fig. 2b). Yet, for both of these factors, the results vary depending on which baseline periods are used to normalize hazard frequency and severity. The predominance of negative values suggests that the average level of policy change has been lower in countries facing more frequent and severe hazard events than countries with fewer and less severe hazards.

Testing the robustness of these results, we first conducted correlation analyses for all measures, including separate analyses of policy change in relation to different combinations of hazard event types (Supplementary Table 2). Results confirmed that the relationship between hazard events and policy change is very weak and slightly negative across hazard frequency and the three severity measures. Only a few significant differences were found across income levels and hazard event types (results based on 30-year baseline). Next, we controlled for cross-country variance in HFA starting values (i.e., that countries reported different initial progress scores in any of the first two HFA evaluation cycles) by calculating the average PFA change ratio between 2011–2015 and 2007–2011 over the average PFA change in 2007–2011 (Eq. (3), "Methods"). Analyses with the normalized PFA measure did not generate different results compared to average PFA changes (Fig. 1 and Supplementary Fig. 2) and PFA difference scores (Fig. 2 and Supplementary Fig. 3). Thus, we find that differences in initial PFA scores did not influence the relationship between natural hazard exposure and policy change within the study period.

**Country cases**. Although the study shows that no relationship exists between natural hazard events and DRR policy change in our sample of 85 countries, we can still observe considerable differences when comparing across individual countries. These differences provide opportunities to strategically select countries for in-depth comparisons. Different selection logics are possible, and Fig. 3 exemplifies two of the alternatives. To reiterate, since hazard severity and frequency depend on countries' previous

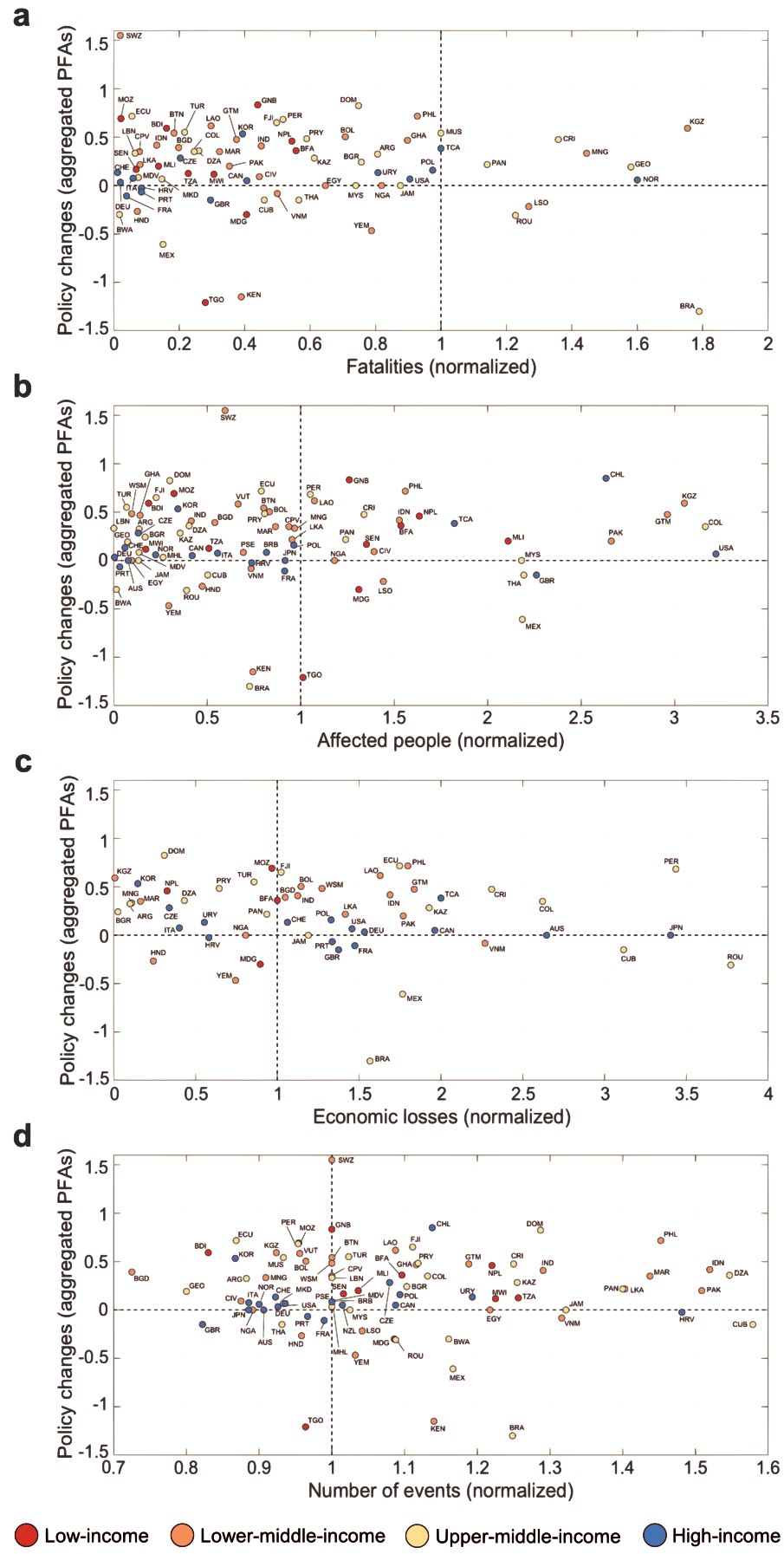

**Fig. 1 Average aggregated policy changes in relation to natural hazard event frequency and severity measures.** Plots demonstrating the relationship between fatalities (**a**), affected people (**b**), economic loss (**c**), number of events (**d**), and average changes in aggregated HFA PFAs by income levels in the World Bank's fiscal year 2015 (red = low income; orange = lower–middle income; yellow = upper–middle income; blue = high income). Country acronyms are provided in the Source Data file. Frequency and severity measures are normalized against the 30-year country baseline. Normalized index ≤1 indicates that hazards are less or equally frequent and severe as the 30-year baseline long-term average. Conversely, normalized index values >1 represent more frequent and severe events than the 30-year baseline. Frequency and severity scales have been shortened for readability, to the effect that some countries are excluded from **a–d**. Countries not shown in **a**: Chile (CHL, normalized fatality score = 2.54), Australia (AUS, 2.94), Japan (JPN, 3.27), and Samoa (WSM, 4.13); **b**: Uruguay (URY, normalized affected people score = 6.26) and Macedonia (MKD, 4.90); **c**: Chile (CHL, normalized economic loss score = 5.47), New Zealand (NZL, 7.68), Thailand (THA, 8.06), and Malaysia (MYS, 5.18); **d**: Turks and Caicos Islands (TCA, normalized number of events score = 2.0). Source Data are provided as a Source data file.

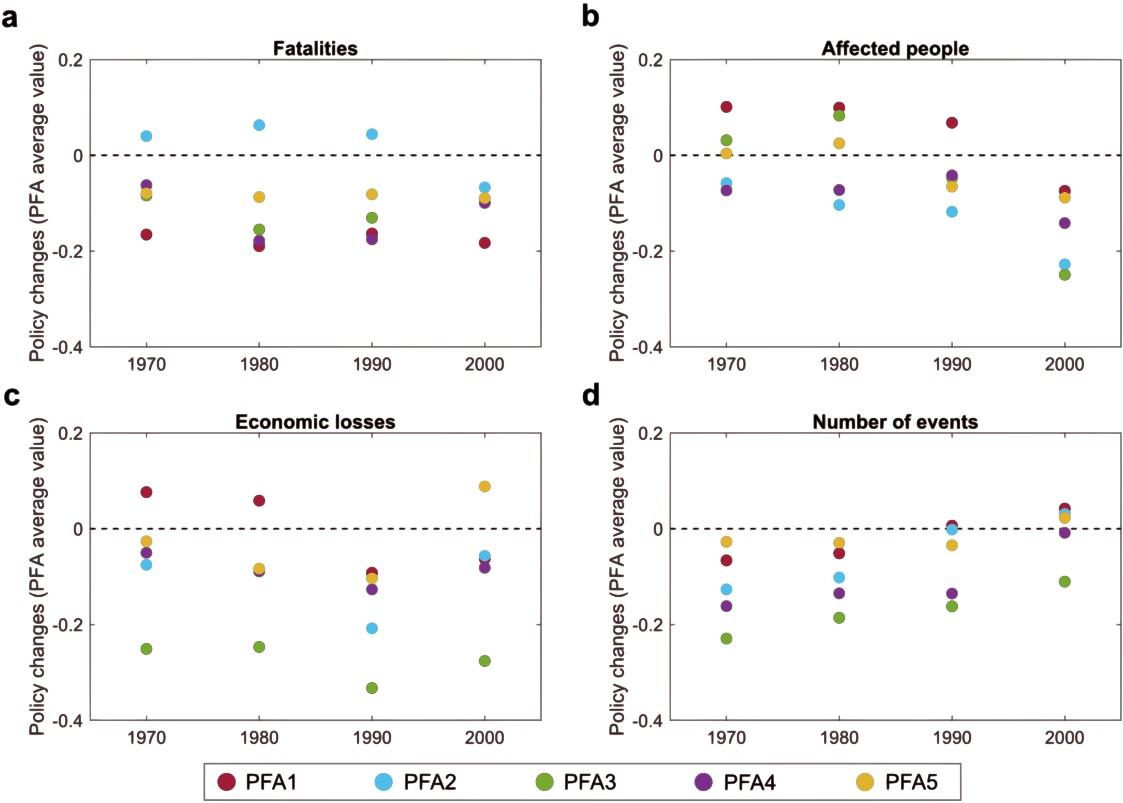

**Fig. 2 Differences in policy changes in relation to levels of hazard event severity and frequency.** Plot comparing changes in average values of the Hyogo Framework for Action Priority for Action areas (PFA), indicating self-reported achievement measures for enhancing disaster resilience. Policy changes are displayed for fatalities (**a**), affected people (**b**), economic losses (**c**), and frequency (**d**) and are estimated as the difference of the average PFA values of the countries that experienced normalized hazard measures >1 (higher frequency/intensity than in the four baseline periods) and the average PFA values of the countries in which the normalized hazard measures is <1 (lower frequency/intensity than in the four baseline periods). Source Data are provided as a Source data file.

experience, we base the comparisons on normalized measures in relation to long-term country averages (30-year baseline) and not in absolute terms.

In a "type *a*" comparative logic, we identify pairs of countries that are consistent with the aggregated pattern that hazard frequency and severity do not influence DRR policy change. These are pairs of countries with different hazard exposure levels that resulted in no change or similar levels of policy change. In contrast, a "type *b*" comparative logic entails two countries with the same level of hazard exposure but different levels of policy change toward enhanced DRR capacity (i.e., positive policy change scores). Hence, in a type *b* case comparison, one case is consistent with the aggregated pattern reported in this study (high hazard event frequency or severity followed by policy stability), whereas the other case (high frequency or severity

followed by significant policy change) is consistent with the disaster reform hypothesis.

We start by identifying two "type *a*" country pairs that illustrate the aggregated pattern that DRR policy change is unassociated with hazard severity and frequency. Canada and Australia constitute one illustrative high-income pair in relation to fatalities (Fig. 1a). In our data, Canada and Australia differed substantially in the level of hazard fatalities during the exposure period, yet they still reported similar levels of DRR policy change (Fig. 3). Specifically, the EM-DAT data show that Canada suffered a total of 15 fatalities due to floods and storms, for a normalized fatality score of 0.41, i.e., below the long-term country average in the study period. Australia, while also facing storms and floods, suffered 21 fatalities, for a fatality score of 2.94, which is above the long-term country average. Despite these differences

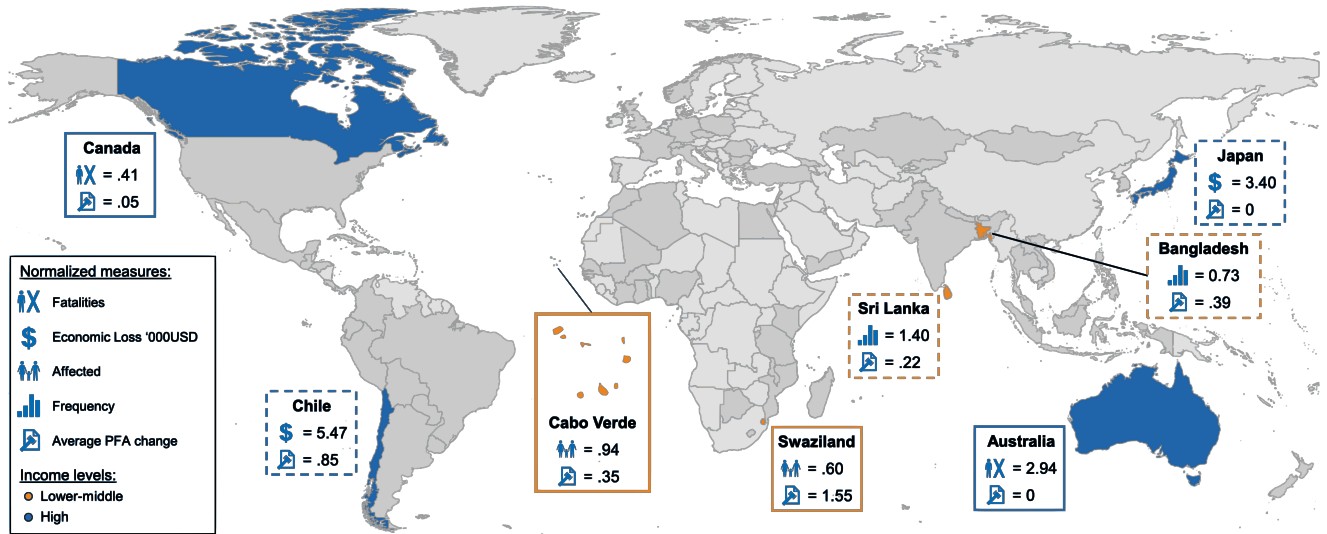

**Fig. 3 Map of candidate countries illustrating different selection logics involving natural hazard frequency, severity, and policy change.** Country cases are identified based on the results shown in Fig. 1a–d. Each pair includes two countries at the same income level (World Bank's fiscal year 2015) experiencing similar hazard impacts (normalized measures) but different levels of average PFA changes (Cabo Verde and Swaziland, lower–middle-income category [orange]; Chile and Japan, high-income category [blue, dashed line]), and two countries experiencing different hazard impacts but similar levels of average PFA changes (Canada+Australia, high-income category [blue]; Bangladesh+Sri Lanka, lower–middle-income category [orange, dashed line]). Frequency and severity measures are normalized in relation to the 30-year baseline. All other countries ($n = 77$) included in the study are shaded (dark gray). Source Data are provided as a Source data file. World map created using a dataset in shapefile format downloaded (November 12, 2020) from http://thematicmapping.org/downloads/world_borders.php under the Creative Commons Licence Attribution-ShareAlike 3.0 Unported (CC BY-SA 3.0): https://creativecommons.org/licenses/by-sa/3.0/ (no modifications made to the material). Icons source: OCHA (United Nations for the Coordination of Humanitarian Affairs), downloaded (November 12, 2020) for free on ReliefWeb: https://reliefweb.int/report/world/world-humanitarian-and-country-icons-2012.

in fatalities, DRR policies in both countries remained unchanged (policy change score for Canada = 0.05 and Australia = 0).

Sri Lanka and Bangladesh represent a similar pattern in relation to event frequency (Fig. 1d). Sri Lanka and Bangladesh—both classified by the World Bank as lower–middle-income countries, also located in the same region—experienced different levels of hazard frequency; whereas Sri Lanka recorded a number of events exceeding the long-term average (score = 1.40), the number of events in Bangladesh fell below the long-term average (score = 0.73). However, the reported policy changes in these countries were still relatively similar (Bangladesh = 0.39; Sri Lanka = 0.22).

In our data, we can also identify "type b" country pairs with similar experiences of hazard severity but with different levels of DRR policy change. In relation to affected people (see Fig. 1b), one example is Swaziland and Cabo Verde, which were both classified as lower–middle-income countries within the study period. Both of these countries experienced hazard events affecting fewer people compared to the average number of people affected by hazards in the period 1980–2011 (30-year baseline). In Swaziland, 414,400 individuals were affected (0.60 in relation to the long-term average), and in Cabo Verde 21,297 individuals were affected (0.94 in relation to the long-term average). Nevertheless, despite these relatively similar impacts, Swaziland and Cabo Verde reported different levels of policy change; Cabo Verde was close to policy stability, (policy change = 0.35) while Swaziland had the highest (= 1.55) policy change value of all countries in the dataset. It can also be noted here that normalized frequency (Fig. 1d and Supplementary Fig. 3) was identical for these two cases (1.0), which suggests that differences in policy change cannot be attributed to differences in event frequency.

Another comparison concerning the potential policy impacts of economic losses (Fig. 1c) is between Chile and Japan (high income), which both experienced several major hazard events causing economic losses significantly exceeding the 30-year average (Japan 3.40, Chile 5.47). In Japan, this included the 2011 Tōhoku earthquake and tsunami, causing the Fukushima Daiichi nuclear plant disaster. Despite this, Japan reported no policy change in the subsequent HFA evaluation cycles (policy change score = 0). In contrast, Chile reported the second highest policy change score (0.85) of all countries in the dataset.

Figure 3 displays candidate cases that can be further studied in depth to identify within-case factors and processes explaining why we see varying policy impacts across countries after natural hazard events. Type b comparisons are particularly interesting as they suggest, contrary to the general pattern reported in this study, that natural hazard severity and frequency may still drive DRR policy change in some cases. The data reviewed here also suggest that several relatively stable factors, including income levels, geographic location, and hazard types, do not appear to matter in driving DRR policy change after natural hazard events. However, more work is warranted to examine the potential influence these and other factors have on DRR policymaking after natural hazard events.

## Discussion

Advancing knowledge of whether and how natural hazard events affect countries' efforts to improve DRR policy is essential for reducing vulnerability and disaster losses. Here we show that, during the time span of the HFA regime (2007–2015), natural hazard event frequency, fatalities, economic losses, and affected people have not influenced policy change toward improved DRR globally. This finding provides direction for future research and has practical policymaking implications for achieving DRR around the world.

Two competing views exist in the scientific literature concerning the relationship between event frequency and policy

change. From one viewpoint, exposure to repeated hazard events is predicted to constrain policy change because policy-makers have to devote limited resources to recovery efforts at the expense of developing proactive DRR policy[19,43]. In this scenario, sometimes labeled the "tyranny of the urgent," the need for rapid reconstruction will override participatory efforts targeted at sustainable development[44], which in turn may reduce the ability to cope with the next event[45]. A competing viewpoint, emphasizing policy-learning, posits that exposure to repeated hazard events leads to an aggregation of experience through time, which has the potential to gradually alter policy-makers' beliefs and culminate in major policy change[37,46]. Other findings indicate a positive relationship between event frequency and economic growth, implying an adaptation effect[47]. The results of this study cast some doubt over the latter viewpoint, since we found no correlation between hazard frequency and policy change (Fig. 2 and Supplementary information Table 2). Notwithstanding this finding, it is important to note that our analysis does not suggest that DRR policy stability can be attributed to a continuous focus on relief and reconstruction as intimated by some studies[19,20].

Our results are consistent with insights about institutional inertia, which predict that enduring policy legacies sustain rigid beliefs and practices that prevent policy renewal. These accounts depict natural hazards as potential focusing events whose impact on policymaking is ultimately mediated by a range of factors. Disruptive natural hazards are often followed by a surge in public attention, yet whether episodes of heightened attention actually enable policy change depends on the degree of polarization of the policy domain, which affects the prospects for group mobilization, efforts of policy actors to frame events in ways that support their beliefs and preferences, and their access to policymaking venues[8]. How such processes unfold clearly varies and may be driven by media propagation or take place in expert-dominated domains away from the media spotlight[48–51].

Other trajectories of policy change after natural hazards involve overreactions and symbolic politics in response to institutional crises. This perspective recognizes that severe hazard events sometimes trigger crises with "frame-breaking" qualities where dramatic imagery of widespread human suffering and material destruction come to symbolize neglected risks and vulnerabilities and, ultimately, the inability of the state to protect its citizens. During such "institutional crises" when the media, the general public, political opponents, and other stakeholders publicly question the effectiveness and legitimacy of established policies, policy-makers may either set out to demonstrate a commitment to learning and reform or seek to provide reassurance that the existing system is robust as it stands[52]. Also, for policy-makers, the pressure to "do something" can create perverse incentives to pursue hasty, and even disproportionate, policy responses as a means to demonstrate political determination in the face of public arousal and criticism[53].

Theoretically, pathways of policy change after hazard events are predicated on different levels of democracy. Established theoretical accounts of policymaking in the wake of hazards and crises may, thus, have limited explanatory leverage in less democratic countries where accountability, agenda-setting, and stakeholder mobilization is constrained by political repression, limited opposition, and fragile political institutions[54]. Several countries identified in this study with relatively high DRR policy change scores (average PFA change above 0.5, Fig. 1)—for example, the Philippines, Guinea Bissau, and Burundi—generally score low on political rights, civil liberties, and other attributes of democratic governance[55]. Cases of hazard-induced policy change in less democratic countries, hence, provide opportunities to elaborate explanations of hazard-driven policymaking under challenging political conditions. Experiences from Nepal[56] and Mozambique[43], for example, suggest that the active involvement of non-governmental organizations and multilateral agencies and donors can help overcome departmental protectionism and other barriers to improved DRR policies in countries that lack stable democratic institutions. Whether and how such organizations enable countries' efforts to improve DRR policy after extreme events is a critical topic for future research[17,57].

This study also contributes to research on the adaptation deficit, which suggests that due to poor adaptive capacity, low-income countries are generally more vulnerable to climate extremes than high-income countries[39–41]. While this study does not measure vulnerability (see "Methods"), it shows that DRR policy change, which represents one crucial dimension of adaptation[58,59], varies marginally across income levels (Fig. 1) and that stability is more common in high-income countries than in low-income countries. The study also suggests that hazard event frequency and severity do not have different effects on policy change across income levels (Supplementary Table 2). We hereby conclude that, when adaptation is measured as DRR policy change after periods of natural hazard events, we do not find any empirical support that low-income countries would have less adaptive capacity than high-income countries.

There is a need for further research to unveil mechanisms enabling policy change after extreme events[12,22,48]. Our study informs this work in two important ways. First, our results indicate that changes in DRR policy reported by countries do not depend on how many natural hazard events countries have been exposed to or the severity of those events. This finding holds regardless of the hazard event type, which suggests that the likelihood of policy change after extreme events does not depend on the type of hazard. The study hereby supports the rather straightforward insight in the literature that factors associated with the policy process have to be taken into account to explain policy change after natural hazard events[48,60]. Future work investigating these factors should recognize that policy change evolves through different trajectories, i.e., that different situations give rise to different socio-political dynamics, all of which may result in policy change. This is consistent with the call from the Global Sustainable Development Report[61] for policy-relevant knowledge about transformation pathways in different settings. Second, we provide an approach to strategic sampling of country cases that either conform to or deviate from the general pattern that natural hazard events are unassociated with DRR policy change. These country pairs (Fig. 3) serve as a guide for future in-depth comparative case studies to unveil drivers of policy change in different settings.

Despite progress in DRR policy adoption globally under the HFA regime, the prevalence and severity of natural hazards are increasing around the world. In response, the UNDRR is exhorting governments to exploit natural hazards for lesson-drawing and enhancing DRR policy[62]. We find here that this work has lagged behind. One practical implication of this study is that many natural hazard events are unexploited opportunities for learning and policy reform to strengthening DRR and, ultimately, achieving the SDGs. Therefore, it is also crucial to investigate whether and how countries can arrive at improved DRR policies and capacity building without directly suffering the devastating consequences of a major natural hazard event. Insights about such vicarious learning can be achieved by studying exchanges of experience and best practices across countries through policy diffusion or transfer[25,63] enabled by collaboration across sectors and local participation[64].

## Methods
### Data sources
*Natural Hazards Database.* We retrieved natural hazard impact data from the EM-DAT global dataset[35] (see "Data availability," data downloaded March 8, 2019) for

the period from 1970 to 2011, including a total of 10,976 individual natural hazard events. The EM-DAT database includes information about natural hazard events causing at least 10 fatalities, 100 affected people, a call for international humanitarian assistance, or a declaration of a state of emergency. Here we collected data for (i) the total number of hazard events, (ii) natural hazard fatalities, which is the sum of missing people and lives lost due to a hazard event, (iii) total people affected, which represents the sum of individuals requiring immediate assistance (injured and homeless requiring basic survival needs) after a natural hazard, and (iv) economic damage, representing damage to property, crops, and livestock. For this analysis, we included events from all six natural hazard categories: geophysical (earthquake, mass movement, volcanic activity), meteorological (extreme temperature, fog, storm), hydrological (flood, landslide, wave action), climatological (drought, glacial lake outburst, wildfire), biological (epidemic, insect infestation, animal accident), and extraterrestrial (impact, space weather).

*DRR policy database.* We collected data on DRR policy from UNISDR evaluations of the HFA. From the HFA National Progress Query Tool (see "Data availability," data downloaded May 2, 2018), we retrieved data for all four evaluation cycles of the HFA regime: 2007–2009 (number of available country reports = 61); 2009–2011 ($n$ = 133); 2011–2013 ($n$ = 113); and 2013–2015 ($n$ = 95). These data entail self-reported measures of countries' level of progress (so-called "progress scores")—on a scale from 1 (minor achievement) to 5 (comprehensive achievement)—in implementing 22 policy goals within five PFA areas (see Supplementary Table 1).

*Income-level data.* For income levels, we used World Bank data, updated July 1, 2017 (see "Data availability," data downloaded March 20, 2018), using data for income levels in the year 2015. If not otherwise stated, classifications of income levels have remained stable across the study period.

### Data limitations
*DRR policy data.* Several caveats apply regarding the quality of the HFA data. The HFA progress scores are self-reported and, therefore, subjective. Responses are provided by different actors in different countries, most commonly by representatives of public agencies in disaster planning and management, and sometimes in collaboration with other stakeholders (representing, e.g., research and civil society). Self-assessment data must always be treated with caution due to potential differences in how countries may have interpreted the indicators, under- or over-reporting, and the possibility of biases[65,66]. To minimize these problems, the UNISDR provided guidance for self-assessment[67], an online tool—the HFA Monitor—for self-assessment monitoring and reporting, a peer-review process, and complementary tools to measure progress such as the Global Assessment Report and reports to the global and regional platforms for DRR[66]. While the data lack an external verification process, because the HFA is voluntary and states are not rewarded or punished for their performance, this may increase its reliability and reduce perverse incentives to inflate progress or for countries to present themselves in the best possible light due to social desirability bias. Moreover, since the HFA scores are not used for resource allocation, there is also little incentive to artificially downgrade performance. Despite some important limitations, the data provided by states about their DRR performance over time provides the richest single storehouse of global information about DRR measures and enables the systematic examination of DRR action. This wealth of information provides valuable insights into the progress of countries' DRR capabilities[68].

*Policy change and vulnerability reduction.* It should be kept in mind that the HFA data do not directly measure vulnerability reduction, which was the goal of the HFA and one of the SDG targets. Investigating whether improvements in DRR policy actually lead to vulnerability reduction is thus not possible with these data and, therefore, an essential next step in this research[18,25].

*National versus local- and regional-level policy action.* Our results concerning national-level DRR policies should be contrasted with the potential effects of hazard events on local and regional level policy action[69]. Studies show that transformations after natural hazards often take place locally, emanating from adaptive efforts by households and civil society groups without the involvement of public actors[17]. External evaluations of the HFA framework, however, suggest that the regime produced limited local impact[70]. Although local policy responses are indirectly accounted for in our study (local priorities and risk assessments are acknowledged within PFA1 and PFA2, see Supplementary Table 1), this is an area where more work is warranted. Regional-level policymaking can also be studied in relation to hazard events. Useful reference cases include, e.g., the development of sustainable flood prevention policy within the European Union in the wake of catastrophic floods in 2002[71] and the development of the Indian Ocean Tsunami Warning and Mitigation System after the 2004 tsunami[72].

*Time frame.* The time period covered here excludes policy changes undertaken prior to the first HFA evaluation cycle in 2007–2009. Many countries adopted policies before the HFA was initiated to reduce disaster risks with a focus on prevention. Therefore, it is possible that some countries entered the HFA

evaluation process at a relatively high level of progress, reducing the space available for improving policy. Nevertheless, we show here (Supplementary Figs. 2 and 3) that there is no significant difference in hazard-driven policy change between countries with different HFA starting values. In other words, countries reporting higher PFA scores in the first two HFA evaluation cycles were neither more nor less prone to report higher scores in the last two evaluation cycles compared with countries that reported lower scores at the outset.

*Natural hazard events data.* The EM-DAT database, although it is one of the world's most comprehensive disaster databases and a recent study showed that its data were quite consistent with the insurance group Munich RE's NatCatSERVICE database[73], is subject to some limitations. There is some missing information, and it is constrained by known inconsistencies in data collection due to improved loss reporting, exclusion of small-scale events[74], and spatial discrepancies resulting from changes in political boundaries[75]. One step to compensate for these limitations in future research is to exploit other global disaster databases to validate the results reported in this study. Data sources on specific hazard types—for instance, the *Global Runoff Database* on floods, the *International Best Track Archive for Climate Stewardship* (IBTrACTS) on tropical storms, and the *Global SPEI* database on droughts—can be consulted for this purpose.

**Description of the methodology.** To empirically test the disaster reform hypothesis, we conducted an exploratory analysis by comparing average policy changes with hazard frequency (measured by the number of events per country) and severity (measures of fatalities, people affected, and economic losses). Our methodology builds upon the steps detailed below.

*Estimation of change in hazard frequency and severity.* To ensure comparability between countries, frequency and the three severity measures were normalized for each country by dividing the averages in 2007–2011 (period $t_1$, Supplementary Fig. 1) with long-term averages over four alternate baseline periods: 40 years (from 1970 to 2011), 30 years (1980–2011), 20 years (1990–2011), and 10 years (2000–2011). For example, in the baseline period of 40 years, the normalized hazard measures are indicated as $\hat{E}^{40y}$, $\hat{F}^{40y}$, $\hat{A}^{40y}$, and $\hat{L}^{40y}$ for event frequency, fatalities, affected, and economic losses, respectively (Figs. 1 and 2). One common approach is to use normalized gross domestic product to normalize country-level economic losses and population to normalize fatalities and affected people[6,73,76]. This approach, however, would not make the results comparable since events in some countries are normally more damaging than events in other countries. Thus this normalization approach could not capture the exceptionality of events that, according to the disaster reform hypothesis, are assumed to trigger policy change.

We, therefore, applied an alternative approach, which considers the frequency and severity of natural hazards relative to each country's long-term averages. Our study, thus, recognizes that hazard impacts are context dependent, i.e., that the frequency and severity of events are relative to the historical experience of each individual country. For example, normalized hazard measures >1 indicate that countries in the 2007–2011 period experienced more severe and/or more frequent hazards than average over the baseline period. Utilizing this approach, we analyzed the data in relation to four baselines periods as benchmarks to test the sensitivity of the statistical results for different hazard events occurring in a period spanning four decades. Figure 2 compares results across all four baselines to discern potential effects from using different time periods that normalize each measure. In other parts of the study (Figs. 1 and 3), we present results based on the 30-year baseline, which represents a reasonable compromise between excluding older events data with potential reliability issues[75] and ensuring a sufficiently large sample of historical events.

*Estimation of policy change.* To generate DRR policy change scores for each country and measurement period, we first calculated the difference between the average of the 22 core indicators within the five PFA areas as:

$$\Delta \text{PFA}_c^p = \overline{\text{PFA}}_c^p(t_2) - \overline{\text{PFA}}_c^p(t_1), \quad (1)$$

where $p$ is each specific PFA area, $c$ is the country, and $t_1$ and $t_2$ are the four HFA evaluation cycles divided into two periods: 2007–2011 (evaluation cycles 1 and 2 combined) and 2011–2015 (evaluation cycles 3 and 4). Aggregated PFA values were calculated as the average value of policy goals in the HFA evaluation cycle periods 2007–2009 and 2009–2011 for $t_1$ and periods 2011–2013 and 2013–2015 for $t_2$. Next, we estimated the aggregated change of PFAs for each country (Fig. 1) as:

$$\Delta \text{PFA}_c = \frac{1}{5} \sum_{p=1}^{5} \Delta \text{PFA}_c^p. \quad (2)$$

Averaging was employed to cope with the fact that not all countries submitted national progress reports for each evaluation cycle. Thus, if a country completed only one evaluation cycle, we calculated the PFA value based on the scores from that cycle. In order to filter the influence of the PFA value in the evaluation cycles

periods 2007–2011, we calculated a normalized PFA change as:

$$\overline{\Delta\mathrm{PFA}_c^p} = \frac{\overline{\mathrm{PFA}_c^p}(t_2) - \overline{\mathrm{PFA}_c^p}(t_1)}{\overline{\mathrm{PFA}_c^p}(t_1)}. \qquad (3)$$

*Analysis.* Once policy change scores and hazard measures were calculated for the different periods, we filtered the countries that were in both datasets to compile a unique, consistent dataset. The dataset included countries that: (a) completed at least one of the two evaluation cycles in the first HFA period ($t_1$) and one cycle in the second period ($t_2$) ($n = 94$) and (b) experienced hazard events in the 2007–2011 period ($n = 85$). Nine countries (Anguilla, Armenia, Bahrain, British Virgin Islands, Finland, Monaco, Saint Kitts and Nevis, Seychelles, and Sweden) with valid PFA change scores, but no recorded hazard events within the exposure period, were excluded for the final sample of 85 countries. Due to missing data in the EM-DAT database, the sample size differs regarding hazard event frequency ($n = 85$), fatalities ($n = 81$), people affected ($n = 84$), and economic damage ($n = 59$). Then, in addition to aggregating PFA scores combining all PFAs (Eq. (2) and Fig. 1), we also considered separated scores for the five PFAs and aggregated the results over the 85 countries (Fig. 2). We divided the countries in the sample into two sub-samples representing, first, countries that experienced higher hazard frequency/intensity than the baseline period (normalized hazard measures >1), and second, countries that experienced lower frequency/intensity (normalized hazard measures <1). We then calculated average values over the countries belonging in the sub-samples for the five PFAs as:

$$\Delta\mathrm{PFA}^p = \frac{1}{N}\sum_{c=1}^{N}\Delta\mathrm{PFA}_c^p\bigg|_{H>1} - \frac{1}{M}\sum_{c=1}^{M}\Delta\mathrm{PFA}_c^p\bigg|_{H<1}, \qquad (4)$$

where $H$ is normalized hazard measures and $N$ and $M$ are the total number of countries in the subsamples of $H > 1$ and $H < 1$, respectively. Given the exploratory nature of our study, we used a Mann–Kendall non-parametric statistical trend test to calculate the Theil–Sen estimator and $p$ values between each normalized severity index and average PFA changes for the different baseline periods (Supplementary Table 2). All Theil–Sen estimators were within the range −1.19 to 0.34, with an average value of −0.08 and a standard deviation of 0.27, and only six of them were statistically significant ($p \leq 0.05$). It is worth noting that the highest variability of the Theil–Sen estimator was found when analyzing the number of events, while the smallest variation was observed for the affected people. One limitation of this method is that there might be other (confounding) variables that are potentially obfuscating the result but are not accounted for in our analysis. This suggests that future studies should explore more complex relationships with econometric or causal inference methods.

**Case study method for validating country DRR progress scores**. We conducted an in-depth examination of two cases to investigate whether the HFA scores and the score changes between the HFA evaluation cycles were consistent with how DRR work in the selected cases had been described in the scientific literature. The examination procedure consisted of five steps.

First, we selected two cases with the highest PFA change scores in the dataset: Swaziland (average PFA change score = 1.55) and Chile (average PFA change score = 0.85). The procedure for estimating policy change is detailed in "Methods" (Eqs. (1) and (2)), and final PFA change scores for all countries are reported in the Source data file.

Second, we identified the PFA core indicators (Supplementary information, Table 1) that changed the most between the two periods in each country, respectively. As specified in Eq. (1), these changes were calculated based on the differences in average PFA progress scores in the first two HFA evaluation cycles and the last two cycles. However, Swaziland and Chile submitted reports in two of the four evaluation cycles, which means that the estimation of policy change was based on two evaluation cycles in both cases: in Swaziland 2007–2009 [cycle 1] → 2011–2013 [cycle 3], in Chile 2009–2011 [cycle 2] → 2011–2013 [cycle 3]. The Swaziland case included four core indicators that changed from a reported progress score of 1 in the first period to 4 in the second period. In Chile, the two core indicators that changed the most—from 2 to 4 and from 2 to 5, respectively—were included.

Third, we reviewed the substantive qualitative descriptions provided by each country in the HFA reports to justify the reported scores for each core indicator. By doing so, we were able to access more detailed information about concrete actions taken (or not taken) that were cited by the reports as justifications for the scores given. These descriptions were then compared with evidence reported in the scientific literature.

Fourth, to find relevant literature, we searched for previous studies addressing the issues covered by each core indicator. Specifically, we searched Google Scholar (searches conducted between May 4 and June 5, 2020) using different search strings and reviewed the first 50 studies listed. For example, to identify case studies of issues associated with core indicator 1.4 (A national multi-sectoral platform for DRR is functioning) in Swaziland, we used the following search string: "Swaziland+disaster risk reduction+platform." In some instances, we also replaced key terms with related concepts (e.g., "coordination" instead of "platform," and "DRR" instead of "disaster risk reduction"). In this step, we excluded studies that made references to the HFA reports as a data source. Peer-reviewed journal articles, book chapters, Ph.D. dissertations, and published research reports were included.

In the last step, we compared the qualitative descriptions from the HFA reports with accounts derived from the literature to establish whether these were consistent. As a way to minimize the risk of confirmation bias, we searched for evidence in the literature that both corroborated the descriptions and evidence that pointed toward an alternative interpretation. The results of these validation tests are detailed in the Supplementary information (section 3) along with references to the literature used.

**Reporting summary**. Further information on research design is available in the Nature Research Reporting Summary linked to this article.

## Data availability

Natural hazard events data, including the number of events, fatalities, economic loss, and people affected, are accessible through the International Disaster Database (EM-DAT), administered by the Centre for Research on the Epidemiology of Disasters (CRED), https://www.emdat.be. EM-DAT data are freely available but accessible by a data request. Data on national-level DRR progress can be freely downloaded through the Hyogo Framework for Action (HFA) National Progress Query Tool, https://www.preventionweb.net/applications/hfa/qbnhfa/home. Data on country income levels underlying Figs. 1 and 3 were derived from the World Bank, https://datahelpdesk.worldbank.org/knowledgebase/articles/906519-world-bank-country-and-lending-groups. All analyses in this work were performed with the MATLAB software version R2019b. In particular, the statistical analyses were carried out by using the MATLAB function ktaub available at: https://se.mathworks.com/matlabcentral/fileexchange/11190-mann-kendall-tau-b-with-sen-s-method-enhanced. Source data are provided with this paper.

## Code availability

The custom code and mathematical algorithm generated for this study have been deposited in the following public repository: https://www.statsvet.uu.se/research/trampoline/data-repository/.

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

## Acknowledgements

We thank the International Disaster Database (EM-DAT) team for granting access to natural hazard events data, Beatriz Quesada Montano for assistance with initial data preparation, and Jacob Hileman for comments on the main text. This work was

conducted as part of a project entitled "The transformative potential of extreme weather events (TRAMPOLINE)" supported by the Swedish Research Council (grant No. 2018-03977 to all authors). All authors are grateful for the support from the Centre of Natural Hazards and Disaster Science (CNDS). Additional support was received from the Swedish Research Council FORMAS within the ERA-NET project STEEP-Streams (to M. M.) and the European Research Council (ERC), which provided funding for G.D.B.'s project entitled "HydroSocialExtremes: Uncovering the Mutual Shaping of Hydrological Extremes and Society," H2020 Excellent Science, Consolidator Grant No. 771678.

## Author contributions

All authors designed the study. M.M. developed the custom code and conducted data analyses with G.D.B. D.N. was the lead author, C.F.P., G.D.B., and M.M contributed to the writing of the text. All authors contributed to the analysis and interpretation of results, reviewed the manuscript, and gave approval for publication.

## Funding

## Competing interests

The authors declare no competing interests.
