## [Peer Review File · Nature Communications]

Reviewers' comments:

Reviewer #1 (Remarks to the Author):

Review of the paper "Disaster risk reduction policy change after natural hazard events"

Overall evaluation

The paper and global data examined provides new insights into the topic of whether and how natural hazards at a global scale (with national scale resolution) might trigger or facilitate changes in DRR policies. In this regard, the topic is novel and it provides a new global overview.

However, I am less convinced that the paper at this moment provides very strong evidence for the conclusions derived. Particularly, the limitation of the self-assessment within the HFA reporting and the question which national institution reports such evaluations might have severe influence on the answers and the judgements made.

In addition, the paper could also consider that important policy changes have occurred at the regional rather than at the national scale after extreme events. The flood risk management directive of the EU (even though many high-income countries reported policy stability) and the tsunami regional early warning system in Southeast Asia are just two prominent examples, which also modified DRR policies at national scale in various countries significantly. Perhaps the EU flood risk directive did not change DRR institutions, however, new regulations on how to treat flood risks implied significant changes on how flood risks are viewed, assessed and managed. Also in Southeast a new regional management and information system on tsunami risk reduction was introduced after the Indian Ocean tsunami (actually it took some years to change policies and develop such institutional and technical systems).

In this context, I would suggest that the authors should also try to identify countries where policy changes in DRR in the past 2 decades have occurred and are documented in the scientific literature and explore whether these changes were also reported within the HFA reporting. The country examples provided by the author team are quite interesting, however, the difference is marginal between 15 fatalities in Canada and 21 fatalities in Australia. In this regard, I would recommend using more significant examples in terms of similar hazards – that caused different impacts (fatalities). E.g. the Haiti earthquake in 2010 (magnitude 7,0 Mw) versus the New Zealand/ Christchurch earthquake in 2011 (magnitude 6,3 Mw). While in New Zealand, the number of fatalities due to the earthquake was about 185 the fatalities in Haiti accounted for about 300.000 people. Hazard impacts (societal severity) might be more significant than the hazard frequency for questions of policy changes. Similarly, one could explore whether and how Indonesia and Sri Lanka reported DRR policy changes in the HFA reporting periods also in the context of the Indian Ocean Disaster (tsunami) and after major floods that occurred later - reporting periods (2007-2011 and 2011-2015).

In summary, I would recommend the authors to revise their paper and to focus on country examples that are more significant in terms of different hazard impacts. The global analysis is an interesting entry point, however, its value or validity would have to be juxtaposed with more in-depth country results. The authors have the empirical data to compare case study countries where policy changes are reported in the literature and the policy changes reported in the HFA monitoring process.

More detailed comments:

Figure 1 is interesting, however, figure 2 is very difficult to understand/ read. In addition, as said before the number of events might not be very telling.

It would be helpful to compare and juxtapose the finding that "an increase in hazard frequency and severity is generally followed by less policy change" from the global analysis with well

documented national or regional case studies (considering both cases - DRR policy changes and countries where such changes did not occur after major hazard impacts).

In addition, I would recommend the authors to compare for selected countries policy changes reported within the context of the HFA and those reported in the literature. The HFA self-reporting might have severe limitations particularly within countries with governance challenges.

The discussion of the data limitations on page 11 is good, however, for a better understanding it would be helpful if the hazard types examined and the limitations were briefly mentioned also in the beginning.

In my view the authors should also discuss or derive recommendations for improving DRR reporting within the HFA. For example, it is interesting to note that the second most changed category in the HFA reporting was the "reduction of underlying risk factors". Considering that changes in underlying risk factors particularly require changes not only in DRR policies but also in development strategies and policies it would be interesting to learn more about the institutions that provided the answers. Consequently, more emphasis is needed on how issues how DRR policy changes are reported and whether countries that reported such improvements rank among those that reduced their vulnerability to these hazards (lower fatalities/hazard severity).

Overall, the results are important for the scientific community in the field of DRR, however, as said before, I would be more careful in deriving such a strong conclusion from it. Particularly, since the HFA data is a self-assessment that has to be treated with caution. In addition, the number of hazard events might not be very telling for policy changes. In contrast the severity or hazard impacts are important for mechanisms of international assistance, potential changes in DRR policies and for the evaluation of the effectiveness of existing DRR regimes. In this regard, I was wondering whether figure 1 would change if you calculate the base line just on the basis of the severity of the impacts of hazards.

In summary, I recommend the authors to revise their manuscript.

Joern Birkmann

Reviewer #2 (Remarks to the Author):

Summary

The authors analyze changes in disaster risk policies after natural hazard events in 85 countries around the globe. The main analysis of the manuscript is a correlation between the frequency and severity of the events and policy change after the natural hazard events. The authors report two major findings. The first finding is that hazard frequency/severity and policy change are not correlated with each other. The second finding is the variability of policy progress of countries experiencing similar frequency and severity of hazards. This finding is based on a direct comparison of four country pairs. To my knowledge these findings are novel and would potentially be interesting to other scientists from the domain.

However, the significance of these descriptive findings is very limited due to weaknesses of the methods and limitations of the data sets. For example, as the authors also mention, confounding variables need to be considered to strengthen the results' support of the claims made. Without considering effects from other variables (e.g. the economical development of a country during the period) it is hardly possible to associate a change in policies to hazards only. It is usually problematic to investigate causal relationships from correlations only, not considering all relevant variables increases these problems. Even if the methods would be revisited and improved I doubt that the explanatory power of the results can profoundly be improved due to the data limitations. For example, the self-reporting character of the data sets on policy changes hampers a fair

quantitative comparison between different countries.

The manuscript is mostly clearly written. However, the description of the methods and the research question as well as the conclusions could be more clear. The text contains a few changes in tense. The authors are honest about their findings. The description of the methods lacks information, e.g. for the computation of the results shown in Fig. 2. Detailed results of the statistical analysis are missing and should be added to the supplementary material. The source code would also be helpful for the reproduction of the results.

In general, I feel this manuscript would rather be suitable as a perspective than a research article. The manuscript shows open questions and directions of further research, but falls short on answering the question if DRR policies changes after natural hazard events due to limitations in methods and data.

Broad comments

- The research question and the aim of the manuscript should be formulated more clearly.
- The concept of PFA is not very clearly described in the text.
- Have you done the analysis for groups of countries (e.g. only low-income) or group of hazards (e.g. only meteorological hazards)?
- Detailed statistical results of the correlation analysis (correlation coefficients, p-values) should be added to the supplement.
- From my point of view the limitation to periods is problematic. I know this is due to the limitations of the data sets, but it substantially weakens the significance of the findings. For example it could be possible that hazards happened before 2007 lead to policy actions in 2007-2011, but no actions were taken 2011-2015, since all actions have already been taken. This would end up in negative PFA changes, although actions taken in 2007-2011 were already driven by hazards. Maybe the data sets can be corrected for these effects?
- The method description is a bit convoluted and would benefit from a subdivision in smaller sections. Also details on how the differences in Fig. 2 are calculated should be added.
- Supplementary table 2 is missing. I suppose it is referring to the data table, but it should be stated clearly.

Specific comments

- Fig. 1: I suggest using the same scale for the y-axis
- Fig. 1: Why is the baseline period the 30yr period? This should be mentioned and explained in the Methodology.
- Fig. 1: Hazard indices ≤ 1 and > 1 and PFA changes ≤ 0 and > 0 should be explained in the caption
- Fig. 2: Should this be ≤ 0 and > 0 in the figure caption?
- Fig. 2: Does it make sense to show global averages for the single PFAs? What is the meaning of the baseline period comparison? What is the conclusion?
- Fig. 2: Detailed explanation of how these differences are calculated is missing in the methods
- Figure 3: the reasons and interactions could be interesting, but are not analyzed here, which should be done for such an publication
- Line 129: SD of PFA? Where is the SD displayed?
- Line 320-324: I think this statement is not sufficiently supported by your results.
- Lines 421-426: I suggest removing these sentences, since they do not describe the methods which were used to retrieve the results shown in the manuscript

Response memo, manuscript NCOMMS-19-41406-T

Reviewer 1 (R1)

R1.1. The limitation of the self-assessment within the HFA reporting and the question which national institution reports such evaluations might have severe influence on the answers and the judgements made. I would recommend the authors to compare for selected countries policy changes reported within the context of the HFA and those reported in the literature. The HFA self-reporting might have severe limitations particularly within countries with governance challenges.

Authors' response: We have acted on this useful suggestion from Reviewer 1 by conducting two in-depth case-studies of the two countries with the highest policy change scores: Chile and Swaziland. The objective of the case-studies is to compare the substantive justification for the HFA scores provided by these countries in the HFA reports with how these specific measures have been described in the scientific literature. We elaborate and detail a methodology for this purpose. Through the case-studies, we seek to confirm whether the scores, as well as the policy changes undertaken, are supported by the literature. The methodology and the results of the two case-studies, with appropriate references, are placed in the Supplementary information (section 3) and cited in the main paper (lines 545-583). We have also expanded our discussion regarding the strengths and weaknesses of the self-reported HFA data (Methods, lines 424-439).

R1.2. The paper could also consider that important policy changes have occurred at the regional rather than at the national scale after extreme events. The flood risk management directive of the EU (even though many high-income countries reported policy stability) and the tsunami regional early warning system in Southeast Asia are just two prominent examples.

Authors' response: This is a good suggestion and useful case-illustrations. In response, the revised version (lines 454-457) proposes that more work should be conducted on the regional level (in addition to the local level) focusing, for example, on these two cases.

R1.3. I would suggest that the authors should also try to identify countries where policy changes in DRR in the past 2 decades have occurred and are documented in the scientific literature and explore whether these changes were also reported within the HFA reporting. The country examples provided by the author team are quite interesting, however, the difference is marginal between 15 fatalities in Canada and 21 fatalities in Australia. In this regard, I would recommend using more significant examples in terms of similar hazards – that caused different impacts (fatalities). E.g. the Haiti earthquake in 2010 (magnitude 7,0 Mw) versus the New Zealand/ Christchurch earthquake in 2011 (magnitude 6,3 Mw). While in New Zealand, the number of fatalities due to the earthquake was about 185 the fatalities in Haiti accounted for about 300.000 people. Hazard impacts (societal severity) might be more significant than the hazard frequency for questions of policy changes. Similarly, one could explore whether and how Indonesia

and Sri Lanka reported DRR policy changes in the HFA reporting periods also in the context of the Indian Ocean Disaster (tsunami) and after major floods that occurred later - reporting periods (2007-2011 and 2011-2015).

Authors' response: The first part of this suggestion partially overlaps with and is in line with the comment above (R1.1) regarding the comparison between the HFA scores and the literature. Regarding the second part of this comment, our response is twofold. First, we have clarified throughout the paper (e.g. lines 87-91; 118-123; 226-229) that we are not directly comparing frequency and severity factors (for example fatalities) across countries in absolute numbers but rather by using normalized measures. The paper's main objective is to conduct an exploratory analysis to study the relationship between natural hazard events and DRR policy change on a global scale with the aid of a large-N design. One goal of our study, however, is to elaborate and exemplify a strategy for case-selection based on the results of the analysis. The two new case studies supplied in the Supplementary information provides some additional insights, although the purpose of these is to validate the HFA scores (see the response to comment 1, above). Conducting in-depth case comparisons is the next step in this research, but we do not go beyond the validation cases in this study

Next, as we see it, whether hazard impact has a greater influence on policy change than hazard frequency constitutes an open empirical question, which we investigate in the paper (Fig 1 and Fig 2). We have clarified in the introduction (lines 45-47) and conclusion (lines 312-313) that some studies assume that hazard frequency is a potentially important driver of policy change, which still awaits empirical testing.

R1.4. Figure 2 is very difficult to understand/ read. In addition, as said before the number of events might not be very telling.

Authors' response: In an effort to clarify Fig 2, which we agree was not sufficiently explained in the first version, we have improved the readability (by changing the label on the y-axis, line 158) and elaborated the description of the methodology (lines 531-537). The revised version has developed the argument about event frequency as a potential driver of policy change, using appropriate references to the literature. See the response to comment R1.3.

R1.5. It would be helpful to compare and juxtapose the finding that “an increase in hazard frequency and severity is generally followed by less policy change” from the global analysis with well documented national or regional case studies (considering both cases - DRR policy changes and countries where such changes did not occur after major hazard impacts).

The revised version (lines 250-286) highlights several illustrative cases based on event frequency and the three severity factors. The comparative logic has been elaborated (lines 230-238) to better depict both types of cases indicated by R1, including countries that are consistent with the overall pattern indicating no relationship between hazard events and DRR policy change, and countries that corroborate the disaster reform hypothesis. In addition, the countries used for validating the HFA scores (see R1.1) provide referenced examples of both scenarios requested by Reviewer 1, including higher than average hazard severity leading to major policy

change (case of Chile) and lower than average hazard severity leading to major policy change (Swaziland). We have also provided concrete examples of regional cases, see response above (comment R1.2).

R1.6. The authors should also discuss or derive recommendations for improving DRR reporting within the HFA. For example, it is interesting to note that the second most **changed category in the HFA reporting was the “reduction of underlying risk factors”**. Considering that changes in underlying risk factors particularly require changes not only in DRR policies but also in development strategies and policies it would be interesting to learn more about the institutions that provided the answers. Consequently, more emphasis is needed on how issues how DRR policy changes are reported and whether countries that reported such improvements rank among those that reduced their vulnerability to these hazards (lower fatalities/your hazard severity).

Authors’ response: This is a valid comment to underscore potential practical implications of our study. However, since the HFA was replaced in 2015 with the Sendai Framework for Action, we do not provide any recommendations on how to improve the reporting within HFA.

In response to the second part of the comment, the revised version describes in some detail how DRR progress scores are reported (Methods, lines 421-424). Meanwhile, the question whether improvements in DRR actually leads to reduced vulnerability is beyond the scope of the analysis. Such analysis would also require a different methodological approach to isolate causal effects of specific policy actions, which would substantially increase the complexity of the paper. The paper does acknowledge that the data used in this study cannot adequately capture vulnerability reduction, which thus represents an important next step in this research (Methods, lines 441-444).

R1.7. The number of hazard events might not be very telling for policy changes. In contrast, the severity or hazard impacts are important for mechanisms of international assistance, potential changes in DRR policies and for the evaluation of the effectiveness of existing DRR regimes. In this regard, I was wondering whether figure 1 would change if you calculate the base line just on the basis of the severity of the impacts of hazards.

Authors’ response: As we mentioned in relation to comment 3, the study sets out to test whether DRR policy change is more or less likely in relation to hazard event frequency and severity, which is communicated in the new version of Fig. 1. This figure illustrates the relationship between Average PFA change and frequency as well as the three severity factors, respectively.

Reviewer 2 (R2)

R2.1. confounding variables need to be considered to strengthen the results’ support of the claims made. Without considering effects from other variables (e.g. the economical development of a country during the period) it is hardly possible to associate a change in

policies to hazards only. It is usually problematic to investigate causal relationships from correlations only, not considering all relevant variables increases these problems. Even if the methods would be revisited and improved I doubt that the explanatory power of the results can profoundly be improved due to the data limitations. For example, the self-reporting character of the data sets on policy changes hampers a fair quantitative comparison between different countries.

Authors' response: We agree, this is a very relevant comment. The revised version emphasizes even more clearly that the ambition of the paper is not to present a complete causal model, but to conduct an initial exploratory analysis given the shortage of large-n studies. We explicitly recognize in several parts of the revised paper (e.g. lines 74-75; 540-543) that the exploratory approach is insufficient as a basis for establishing causal relationships. However, the revised version has taken several steps to further enhance the robustness of the findings. First, we have expanded the controls included in the first version (including income-levels (fixed at the status in 2015, based on World Bank data), and different baselines for normalizing hazard frequency and severity) and checked the potential influence of (a) combinations of hazard event types, and (b) differences in starting HFA values. The results of these controls are detailed in the Supplementary information (lines 44-49) and discussed in the main paper (lines 91-92; 207-220). We also more explicitly describe the strengths and weaknesses associated with the self-reported data (under Methods, see response to comment R1.1).

R2.2. The description of the methods lacks information, e.g. for the computation of the results shown in Fig. 2. Detailed results of the statistical analysis are missing and should be added to the supplementary material. The source code would also be helpful for the reproduction of the results.

Authors' response: This is an important point and we agree that including the computational details of the calculation for the results shown in figure 2 will definitely help to enhance the understanding of the study's findings. For this reason, we have undertaken major revisions of the methods-section (lines 486-547) and added a much more detailed approach that includes the equations used to estimate the variables reported in figures 1 and 2, as well as Table 2 and Figures 2 and 3 in the supplementary material. As specified in the instructions included in the letter from the journal editor, we have included the source code and raw input (EM-DAT and policy data) used in our study.

R2.3. The manuscript shows open questions and directions of further research, but falls short on answering the question if DRR policies changes after natural hazard events due to limitations in methods and data.

Authors' response: Following the recommendations of the editor and both referees, we have taken several measures to address limitations associated with methods and data (lines 207-220; 421-439; 486-583; . These steps are depicted elsewhere in this response memo (see responses to comments R1.1, R1.3, R2.2, R2.6, R2.7). In addition, we have made sure to specify throughout that the study can only make statements about policy change as measured by changes in average PFA scores (see e.g. line 77).

R2.4. The research question and the aim of the manuscript should be formulated more clearly.

Authors' response: The revised introduction (lines 71-75) clearly states the aim and the research question of the paper.

R2.5. The concept of PFA is not very clearly described in the text.

Authors' response: The revised version (lines 79-80) succinctly specifies the meaning of the Priority for Action (PFA) areas in relation to the description of the HFA in the introduction.

R2.6. Have you done the analysis for groups of countries (e.g. only low-income) or group of hazards (e.g. only meteorological hazards)?

Authors' response: In the first version of the manuscript, we performed statistical analysis for groups of countries (income-levels) but not for different combinations of hazard event types. We have now done so for hazard event types as well, which we agree is useful for unravelling possible correlations between policy changes and particular hazard event types. For this reason, we have performed additional analyses (Supplementary information, lines 44-49) considering specific combinations of hazard event types, such as floods+droughts, floods+droughts+landslides+storms, floods+droughts+landslides+storms+earthquakes. Results from these analyses did not indicate a strong and significant correlation between policy changes and frequency or severity of natural hazards events. This finding is also discussed in the conclusion (lines 367-369). Hence, we argue (lines 207-220) that these supplementary tests add to the robustness of the main results of the paper.

R2.7. Detailed statistical results of the correlation analysis (correlation coefficients, p-values) should be added to the supplement.

Authors' response: As suggested by the reviewer, we have included detailed statistical results of the correlation analysis in the supplementary material (Supplementary information Table 2, lines 44-49)

R2.8. From my point of view the limitation to periods is problematic. I know this is due to the limitations of the data sets, but it substantially weakens the significance of the findings. For example it could be possible that hazards happened before 2007 lead to policy actions in 2007-2011, but no actions were taken 2011-2015, since all actions have already been taken. This would end up in negative PFA changes, although actions taken in 2007-2011 were already driven by hazards. Maybe the data sets can be corrected for these effects?

Authors' response: This is a valid concern. However, the study's design and the availability of HFA data exclude the possibility of accounting for policy changes undertaken in the 2007-2011 period. This would require data for PFA progress prior to 2007, which is not available. For this reason, the revised version more explicitly acknowledges that the study does not take into account any policy changes prior to 2007 (lines 459-467). In addition, in the revised version (lines 109-113) we have clarified the finding that most countries (n=63, 70%) reported positive policy change, i.e. developed from lower to higher scores between the two periods.

R2.9. The method description is a bit convoluted and would benefit from a subdivision in smaller sections.

Authors' response: The methods section (lines 419-582) has been carefully edited and divided into smaller sections where appropriate.

R2.10. details on how the differences in Fig. 2 are calculated should be added.

Authors' response: As suggested, we have updated the method section and have included the equations (lines 512-547) used to calculate the difference scores shown in figure 2.

R2.11. Supplementary table 2 is missing. I suppose it is referring to the data table, but it should be stated clearly.

Authors' response: This was a typo in the first version. The revised version has been checked thoroughly to ensure that any references to the Supplementary material are correct.

R2.12. Fig. 1: I suggest using the same scale for the y-axis

Authors' response: The same scale (-1.5 – 1.5) is used for all panels in the revised fig 1 (lines 127-131).

R2.13. Fig. 1: Why is the baseline period the 30yr period? This should be mentioned and explained in the Methodology.

Authors' response: We have selected the 30years period as a compromise between a baseline representing a long-time series with adequate sample size (so excluding the 10yr and 20yr period baseline) and a sample including reliable data not affected by old measurements (so excluding the 40yr period). We have included this explanation for why we used this as a baseline in the updated version of the manuscript (lines 507-510).

R2.14 Fig. 1: Hazard indices ≤ 1 and > 1 and PFA changes ≤ 0 and > 0 should be explained in the caption

Authors' response: A description of the normalized hazard index has been added in the caption of figure 1.

R2.15. Fig. 2: Should this be ≤ 0 and > 0 in the figure caption?

Authors' response: We see the misunderstanding that arose from the caption of figure 2. With ≤ 1 and > 1 we were referring to the use of the normalized indices of figure 1 for the calculation of the differences in figure 2. To avoid confusion, we have now modified the caption of figure 2 accordingly.

R2.16. Fig. 2: Does it make sense to show global averages for the single PFAs? What is the meaning of the baseline period comparison? What is the conclusion?

We have taken steps to clarify and simplify the presentation of Fig 2 (see response to comments R1.4, R2.2, R2.10, R2.15). We have added a formulation (lines 171-174) to better

explain the meaning of the baseline comparison. Finally, we have added a new figure to the Supplementary information (Fig. 1, lines 52-57) to explain and visualize the baseline comparison.

R2.16. Fig. 2: Detailed explanation of how these differences are calculated is missing in the methods

Authors' response: Following this comment, we have improved the description of both the caption of figure 2 and the method section, including a clearer step-by-step description of the differences depicted in figure 2. Moreover, we have changed the label of the y-axis to make the figure and related variables easier to understand.

R2.17. Figure 3: the reasons and interactions could be interesting, but are not analyzed here, which should be done for such an publication

We are uncertain what is meant by “reasons and interactions” in relation to Fig. 3. In the revised version, we have clarified the case-selection logic further and settled on four country cases (lines 222-295), shown in Fig. 3. As described above (response to comment R1.1), we also carried out two in-depth case-studies of Chile and Swaziland (which had the highest policy change scores), in an effort to validate the PFA progress scores reported by these countries (Supplementary information, lines 77-236).

R2.18 Line 129: SD of PFA? Where is the SD displayed?

Authors' response: We have calculated the SD of PFA as a SD over the sample of PFA for all the countries and a baseline period of 30yrs. This has been clarified in the revised manuscript.

R2.19. Line 320-324: I think this statement is not sufficiently supported by your results.

Authors' response: We realize that this formulation was a bit blunt. In the revised version (lines 365-369), we have revised the statement to clarify that the finding applies to DRR policy as reported by countries and that this finding is supported by statistical analysis.

R2.20. Lines 421-426: I suggest removing these sentences, since they do not describe the methods which were used to retrieve the results shown in the manuscript

Thank you for pointing this out; as suggested, these formulations have been removed.

Additional changes

In addition to the changes detailed above in response to the reviewers' comments, we have taken several other measures to improve the study:

1. To enhance readability, Figure 1 has been adjusted to display country acronyms. Hereby we do not have to refer readers to the supplementary material (in the previous version the country acronyms were only shown in the supplementary material).

2. We have elaborated the logic underpinning the selection of countries in figure 3 to differentiate between a “type a” logic (cases consistent with the overall pattern identified in the study) and a “type b” logic (where one of the cases is consistent with the disaster reform hypothesis). In doing so, we replaced two of the case-pairs that were included in the first version of the paper (Brazil+Colombia and Togo+Burkina Faso). These pairs were excluded because each one of them included one country with a negative policy change score (Brazil and Togo, respectively), which complicated the presentation of the comparisons. The four cases added include Cabo Verde+Swaziland and Chile+Japan.

REVIEWERS' COMMENTS

Reviewer #1 (Remarks to the Author):

The authors responded to the individual review comments of the 2 reviewers and modified the manuscript significantly.

The changes shown in the manuscript in yellow underscore this revision. Overall, the paper has now a more in-depth argumentation flow and provides additional information in some sections that were particularly discussed within the 2 reviews.

Some questions remain, however, they cannot be answered by the paper and the present ty of HFA report. For example, it is open whether countries within the HFA reporting are doing a good job in terms of reporting policy changes. However, the paper and authors underscores with their quantitative analysis that the frequency and intensity of hazards – also considering now different country groups /World Bank income classes – does not correlate with policy changes reported in the HFA context. Various aspects/comments of the second review (the review from the other person – not from my side) have been considered. The paper provides a useful and innovative contribution to the ongoing discourse about risk governance, risk management and damage and losses.

Reviewer #2 (Remarks to the Author):

Thank you for the revised the manuscript. Although I missed a proper track changes document showing all edits of the manuscript. I still think that the conclusions which can be drawn from the analysis are quite limited. But I appreciate that this is pointed out more clearly in the revised manuscript. The description of the methods is substantially improved as well. In general I think the manuscript contributes to the discussion of global DRR policies and raises important questions. I have two additional minor comments:

Figure 1: The colors of the dots are not explained. Please add a legend and an explanation in the caption.

Supplementary section 3: I appreciated the case study validation very much. Maybe you can add a small summary paragraph at the end of each case study which points out whether and why the PFA-Scores reported are trustworthy. This would enable the reader to get the information faster.

Response memo, manuscript NCOMMS-19-41406A

Reviewer 1 (R1)

The authors responded to the individual review comments of the 2 reviewers and modified the manuscript significantly.

The changes shown in the manuscript in yellow underscore this revision. Overall, the paper has now a more in-depth argumentation flow and provides additional information in some sections that were particularly discussed within the 2 reviews.

Some questions remain, however, they cannot be answered by the paper and the present ty of HFA report. For example, it is open whether countries within the HFA reporting are doing a good job in terms of reporting policy changes. However, the paper and authors underscores with their quantitative analysis that the frequency and intensity of hazards – also considering now different country groups /World Bank income classes – does not correlate with policy changes reported in the HFA context. Various aspects/comments of the second review (the review from the other person – not from my side) have been considered. The paper provides a useful and innovative contribution to the ongoing discourse about risk governance, risk management and damage and losses.

Authors' response: We thank Reviewer 1 for acknowledging improvements in the revised manuscript and for highlighting the remaining question about the HFA reporting, which was also noted by Reviewer 2 (see below). We have partially responded to the HFA reporting issue (whether countries are doing a good job of reporting changes) in the Supplementary material by adding a new summary paragraph for each case-study to discuss the validity of reported policy changes. This has also been clarified in the main text (p. 3).

Reviewer 2 (R2)

Thank you for the revised the manuscript. Although I missed a proper track changes document showing all edits of the manuscript. I still think that the conclusions which can be drawn from the analysis are quite limited. But I appreciate that this is pointed out more clearly in the revised manuscript. The description of the methods is substantially improved as well. In general I think the manuscript contributes to the discussion of global DRR policies and raises important questions.

I have two additional minor comments:

Figure 1: The colors of the dots are not explained. Please add a legend and an explanation in the caption.

Authors' response: We have added a legend under panel d in Figure 1 (p. 5, main text) and an explanation in the figure caption. Thank you for pointing this out.

Supplementary section 3: I appreciated the case study validation very much. Maybe you can add a small summary paragraph at the end of each case study which points out whether and why the PFA-Scores reported are trustworthy. This would enable the reader to get the information faster.

Authors' response: This is a good suggestion. We have added a summary paragraph to each case-study to synthesize the main findings regarding the level (PFA scores) and content (specific policy measures) of policy change in each case respectively.